# Integrated Optimization of Process Planning and Scheduling for Aerospace Complex Component Based on Honey-Bee Mating Algorithm

**Guozhe Yang [1], Qingze Tan [1], Zhiqiang Tian [1], Xingyu Jiang [1,*], Keqiang Chen [1], Yitao Lu [1], Weijun Liu [1] and Peisheng Yuan [2]**

1   School of Mechanical Engineering, Shenyang University of Technology, Shenyang 110870, China
2   School of Mechanical Engineering, Shandong University, Jinan 250061, China; gbevs3v4@163.com
*   Correspondence: xy_jiang9211@sut.edu.cn

**Abstract:** To cope with the problems of poor matching between processing characteristics and manufacturing resources, low production efficiency, and the hard-to-meet dynamic and changeable model requirements in multi-variety and small batch aerospace enterprises, an integrated optimization method of complex component process planning and workshop scheduling for aerospace manufacturing enterprises is proposed. This paper considers the process flexibility of aerospace complex components comprehensively, and an integrated optimization model for the process planning and production scheduling of aerospace complex components is established with the optimization objectives of achieving a minimum makespan, machining time and machining cost. A honey-bee mating optimization algorithm (HBMO) combined with the greedy algorithm was proposed to solve the model. Then, it formulated a four-layer encoding method based on a feature-processing sequence, processing method, and machine tool, a tool was designed, and five worker bee cultivation strategies were designed to effectively solve the problems of infeasible solutions and local optimization when a queen bee mated to a drone. Finally, taking the complex component parts of an aerospace enterprise as an example, the integrated optimization of process planning and workshop scheduling is carried out. The results demonstrate that the proposed model and algorithm can effectively shorten the makespan and machining time, and reduce the machining cost.

**Keywords:** complex aerospace components; process planning; production scheduling; process flexibility; improved HBMO algorithm

## 1. Introduction

With the increasing severity of environmental problems, low-carbon development has become an inevitable choice [1]. There are many resources wasted in multi-variety and small batch aerospace enterprises; complex aerospace components are mixed with small batches, complex geometric features and changeable process routes, which lead to poor matching between processing features and manufacturing resources, and then make production lines switch frequently, resulting in low production efficiency, making it difficult to meet the dynamic and changeable model requirements [2–5]. Process planning and job shop scheduling are two key factors that affect the matching of machining features and manufacturing resources. A large quantity of the literature reveals that integrated process planning and scheduling (IPPS) is advantageous to relieving resource conflict in the job shop, improve equipment utilization and enhance production efficiency [6–8]. As a result, to meet the strict requirements of just-in-time delivery and rapid development in the aerospace industry, research on IPPS is key for aerospace enterprises to realize optimal process decisions, the rapid optimization and matching of manufacturing resources, and effectively improve overall production capacity [9].

The purpose of using IPPS for aerospace complex components is actually to select the appropriate machining method and determine the process sequence for each machining feature of each part under the process constraints; on this basis, the appropriate processing resources are selected for each process, so that the parts can be optimized to a certain extent in many process evaluation indexes such as production efficiency and process cost. For this kind of problem, many works have been carried out on IPPS-related issues from aspects of theoretical research, model construction and algorithm improvement, and have achieved certain results. Mohapatra et al. [10] established an IPPS model with the objectives of minimizing the processing cost, completion time and idle time of machine tools, and then proposed an improved non-dominated-scheduling genetic algorithm to solve the model. Compared with the NSGA-II algorithm, the solution obtained by this algorithm is more excellent in terms of the three objectives. Zhang et al. [11] proposed an object-oriented encoding genetic algorithm to solve the IPPS problem. Xia et al. [12] performed a new dynamic integrated process planning model, which considers the dynamic disturbance of machine tool failure and new task arrival, and adopts the method of combining a hybrid algorithm with rolling window technology to solve the IPPS problem. May [13] and Salido [14] investigated IPPS from the point of view of energy consumption. Zhao et al. [15] built a multi-objective job-shop scheduling model considering alternative process schemes and alternative parallel machine tools, decomposed the job-shop scheduling problem into two sub-problems, namely flexible process route decision and machining process scheduling, and proposed two generations of the Pareto ant colony algorithm to solve it. Mohamad et al. [16] studied the multi-objective job-shop scheduling problem with machine tool processing capacity constraints, built a model with the goal of minimizing the completion time and overtime cost, and proposed a multi-objective genetic algorithm based on ELECTRE to solve it. The proposed algorithm was compared with NSGA-II, SPEA2 and VEGA, and the excellent performance of the proposed algorithm was verified. Chaudhry [17] proposed a genetic algorithm for the IPPS problem which can select the best process planning and job scheduling method in the job shop at the same time. Jin et al. [18] formulated a new MILP model for IPPS in flexible shop floor systems which introduced network diagrams to constrain different operation sequences.

Compared with the traditional optimization algorithm, the intelligent algorithm is widely used to solve the combinational optimization problem, such as IPPS, job-shop scheduling, disassembly scheduling, remanufacturing system scheduling and other problems. Zhang [19] and others studied the distributed integrated process planning problem, used triangular fuzzy numbers to express the machining time of machine tools and the transportation time of parts, proposed a new three-layer encoding method, which improved the genetic algorithm by improving crossover and mutation strategies, and added alternative machine tools and order exchange strategies to enhance the local search ability of the algorithm. Zheng [20] and others studied the job-shop scheduling problem with dual resource constraints, established a model with the goal of minimizing the completion time, combined a knowledge-guided search with a smell-based search, and proposed a new encoding scheme, a knowledge-guided Drosophila optimization algorithm, to solve it. Jiang et al. [21] addressed an energy-efficient scheduling problem of flexible job-shops with complex processes, and they proposed a novel improved-crossover artificial bee colony algorithm. Wang et al. [22] firstly performed an improved genetic algorithm to solve the energy consumption scheduling problem of a remanufacturing system with disassembly, reprocessing and reassembly, and then they provided a new idea for the integrated optimization of a remanufacturing system. Yu et al. [23] presented an improved whale optimization algorithm to solve discrete problems, and the quality and efficiency of the algorithm were verified by experiments. Tian et al. [24] proposed a hybrid optimization algorithm which is a modified discrete gravitational search algorithm to deal with the problem of the emergency scheduling of priority-based rescue vehicles for extinguishing forest fires. Wang et al. [25] addressed a multi-objective invasive weed optimization algorithm proposed to solve the scheduling problem for a remanufacturing system with parallel

disassembly workstations. Tian et al. [26] performed a novel multi-objective bi-population differential artificial bee colony algorithm for the energy-efficient scheduling problem with the multi-resource constraints of a multi-variety and small-batch dynamic flexible job-shop. Yuan et al. [27] proposed a fuzzy disassembly scheduling method based on the fruit fly optimization algorithm, which can effectively solve the disassembly scheduling problem in uncertain environments. Tian et al. [28] presented an improved artificial bee colony algorithm for disassembly scheduling and verified its feasibility and effectiveness. Wang et al. [29] performed a hybrid genetic algorithm based on the variable neighborhood search solution method to solve a remanufacturing system's scheduling problem. Yang et al. [30] addressed a fruit fly optimization algorithm to optimize disassembly line balancing. Tian et al. [31], based on the established stochastic disassembly network graph, combined different disassembly decision-making criteria, and typical stochastic models for disassembly time analysis were developed. Jiang et al. [32] proposed the novel non-dominated sorting genetic algorithm II (NSGA-II) based on adaptive crossover probability and multi-crossover operators to solve the multi-objective optimization model of the laser remanufacturing process. Feng et al. [33] performed a novel hybrid multi-criteria decision-making technique called grey fuzzy TOPSIS.

In summary, the IPPS problem has caused many concerns in the theoretical research, model construction and solution approaches. However, the existing methods are difficult to solve the problem of process route optimization and resource matching of various parts in the production process of aerospace complex components with multiple varieties and models. Additionally, the following problems still need to be solved. First, for the aerospace enterprises where development and production coexist, it is necessary to integrate and optimize process planning and job-shop scheduling for multiple parts at the same time. The existing IPPS models mostly focus on a single workpiece and ignore the utilization of manufacturing resources when multiple parts are produced at the same time, which leads to an unreasonable process route. In addition, due to the large number of parameters and the fact that they cannot be optimized individually in the IPPS model for aerospace complex components, unreasonable encoding will not only make the algorithm slow in terms of search speed and poor in terms of the convergence effect, but also produce a large number of infeasible solutions, which will make it difficult to achieve the expected optimization effect.

Compared to previous studies, the contributions of this paper are summarized as follows. It formulates a multi-objective IPPS optimization model for aerospace complex components with the objectives of achieving the minimum makespan, machining time and machining cost. Moreover, it proposes an improved HBMO algorithm based on four-layer encoding and five worker bee breeding strategies to solve the model. In this way, an efficient IPPS scheme is obtained, the production efficiency is improved and the production cost as well as machine load are reduced. The literature review is summarized in Table 1.

**Table 1.** Literature Review.

| References | Objective Functions | Scheduling Problems | Methodologies |
|---|---|---|---|
| Mohapatra et al. [9] | Makespan, idle time of machines, machining cost | IPPS | NSGA |
| Zhang et al. [10] | Makespan | IPPS | OCGA |
| Xia et al. [11] | Makespan | IPPS | GAVNS |
| Zhao et al. [14] | Makespan, machining cost | IPPS | ACO |
| Mohamad et al. [15] | Makespan, machining cost | IPPS | NSGA (ELECTRE) |
| Chaudhry [16] | / | IPPS | GA |
| **This work** | Makespan, machining time, machining cost | IPPS | HBMO |

The rest of this work is organized as below. Section 2 introduces the mathematical model of the IPPS problem of aerospace complex components. In Section 3, a novel bee

mating algorithm is designed. Comprehensive experiments are conducted in Section 4. Section 5 concludes this work.

## 2. Problem Description

The flexibility of IPPS for multi-variety and small-batch products consists of three aspects: the flexibility of the machining method, the flexibility of the process sequence and the flexibility of machining resources. Generally, a part has multiple machining features, and there are certain machining sequence constraints among different machining features, such as the face before hole. Different machining methods can be selected for each machining feature of parts. Different machining processes can be formed after different machining methods are selected, and each machining process has different optional machining resources, namely candidate machine tools and cutters. In the manufacturing process of parts, the sorting of the feature machining sequence, the selection of a feature machining method and the selection of process resources will have different effects n ther machining cost and machining efficiency. Therefore, the IPPS of aerospace complex components is described as follows: there are several parts $(n_p)$ in the parts to be machined, the $i$ of part $P_i$ has $fea\_num_i$ machining features, the $j$ machining feature $fea_{ij}$ has $method\_num_{ij}$ machining methods, and the $k$ machining method $method_{ijk}$ includes $op\_num_{ijk}$ processes, each of which can be machined on the $n_m$ machine tool of $M_u$ or the $n_t$ tool of $T_v$. Finally, from a global point of view, the sequence of each working procedure of each part is arranged, and the machining machine tools and tools of each working procedure are arranged, so that the completion time, the total running time of the machine tools and the machining cost of all parts are shortened.

The following assumptions need to be met:

(1) The assumption that all parts are produced in the same workshop, and that there is no crossworkshop production;
(2) The assumption that the transfer time, clamping time and preparation time of each working procedure are all included in the processing time of each working procedure;
(3) The assumption that all machine tools are trouble-free and all machine tools are available at zero time;
(4) The assumption that once each working procedure of parts starts machining, it cannot be interrupted.

## 3. Modeling Process

### 3.1. Objective Function

In order to shorten the manufacturing cycle of complex aerospace components, reduce the manufacturing cost and improve the utilization rate of machine tools, this paper constructs a multi-objective integrated optimization model of process planning and shop scheduling for complex aerospace components, aiming at minimizing the makespan, the machining time and the machining cost.

#### 3.1.1. Minimizing the Makespan

To a certain extent, the completion time of processing all parts reflects the processing efficiency of aerospace enterprises. The production set of aerospace enterprises coexists with development and production, and a single piece and small batch are mixed. In the face of strict delivery time requirements, it is necessary to shorten the production time of parts as much as possible. The completion time of aerospace complex components can be regarded as the completion time of the last working procedure of the last part in the parts to be processed, and the calculation formula is as follows:

$$\min\left\{makespan = \max\left(C_{ijkhu} \times Y_{ijkhu} \times X_{ijk}\right)\right\} \tag{1}$$

### 3.1.2. Minimize the Machining Time

To some extent, the total machining time of a machine tool reflects the total machine load for machining all complex components. The total machining time of the machine tool can be regarded as the sum of the machining time of all the processes of the parts to be machined, and the calculation formula is expressed as follows:

$$\min Time = \sum_{i=1}^{np} \sum_{j=1}^{fea\_num_i} \sum_{k=1}^{method\_num_{ij}} \sum_{h=1}^{op\_num_{ijk}} \sum_{u=1}^{n_m} TimeM_{ijkhu} \times Y_{ijkhu} \times X_{ijk} \tag{2}$$

### 3.1.3. Minimize the Machining Cost

In addition to the completion time and the total processing time of the machine tool, the processing cost also needs to be considered. The cost of the machining process of parts includes the use cost of machine tools and tools. The use cost of machine tools or tools is equal to the processing time of each process multiplied by the unit time use cost of machine tools or tools used. Its calculation formula can be expressed as follows:

$$
\begin{aligned}
\min Cost \quad & = CostM + CostT \\
& = \sum_{i=1}^{np} \sum_{j=1}^{fea\_num_i} \sum_{k=1}^{method\_num_{ij}} \sum_{h=1}^{op\_num_{ijk}} \sum_{u=1}^{n_u} TimeM_{ijkhu} \times CostM_u \times Y_{ijkhu} \times X_{ijk} \\
& + \sum_{i=1}^{np} \sum_{j=1}^{fea\_num_i} \sum_{k=1}^{method\_num_{ij}} \sum_{h=1}^{op\_num_{ijk}} \sum_{u=1}^{n_u} \sum_{v=1}^{n_v} TimeM_{ijkhu} \times CostT_v \times Z_{ijkhv} \times Y_{ijkhu} \times X_{ijk}
\end{aligned}
\tag{3}
$$

### 3.2. Constraints

(1)  Only one machining method can be selected for each feature of each part.

Based on the flexibility of machining methods, there are many machining methods for each machining feature of parts, but only one of them is selected in actual machining, and the constraints are as follows:

$$\sum_{k=1}^{method\_num_{ij}} X_{ijk} = 1 \quad i = 1, 2, \ldots, np \quad j = 1, 2, \ldots, fea\_num_i \tag{4}$$

(2)  Only one machine tool can be selected for each process.

Based on the flexibility of manufacturing resources, there is one or more machine tools to choose from in each process, but only one machine tool can be selected in actual processing, and the constraints are as follows:

$$\sum_{u=1}^{n_u} Y_{ijkhu} = 1 \quad h = 1, 2, \ldots, op\_num_{ijk} \tag{5}$$

(3)  Only one tool can be selected for each working procedure.

Based on the flexibility of manufacturing resources, there is one or more tools to choose from in each process, but only one tool can be selected in actual machining, and the constraints are as follows:

$$\sum_{v=1}^{n_v} Z_{ijkhv} = 1 \quad h = 1, 2, \ldots, op\_num_{ijk} \tag{6}$$

(4)  Different processes of the same part cannot be processed at the same time.

At the same time, only a certain feature of the same part can be processed, so there is no parallel processing of the same part process, and the constraints are as follows:

$$C_{ij_1k_1h_1u_1} \times Y_{ij_1k_1h_1u_1} \times X_{ij_1k_1} - C_{ij_2k_2h_2u_2} \times Y_{ij_2k_2h_2u_2} \times X_{ij_2k_2} \geq TimeM_{ij_1k_1h_1u_1} \times Y_{ij_1k_1h_1u_1} \times X_{ij_1k_1} \tag{7}$$

(5)    The same machine tool can only process one part at a time.

At the same time, a machine tool can only process one part, so there is no parallel processing of multiple parts in the same machine tool at the same time. The constraints are as follows:

$$S_{i_1 j_1 k_1 h_1 u} \times Y_{i_1 j_1 k_1 h_1 u} \times X_{i_1 j_1 k_1} - S_{i_2 j_2 k_2 h_2 u} \times Y_{i_2 j_2 k_2 h_2 u} \times X_{i_2 j_2 k_2} \geq TimeM_{i_1 j_1 k_1 h_1 u} \times Y_{i_1 j_1 k_1 h_1 u} \times X_{i_1 j_1 k_1} \tag{8}$$

### 4. Proposed Improved HBMO Algorithm

#### 4.1. Algorithm Design

The integrated optimization of process planning and job-shop scheduling for multi-variety and small-batch aerospace complex components needs to select and combine processing methods and manufacturing resources at the same time, and it is a multi-objective non-linear combinatorial NP-hard optimization problem. Some bionic heuristic algorithms such as the ant colony algorithm and non-dominated-sorting genetic algorithm (NSGA-II) show good adaptability in solving this kind of combinatorial optimization problem. Among them, the ant colony algorithm combines the process of a searching path with the selection of a flexible-process path to obtain the optimal process route. However, the parameter setting of the ant colony algorithm is complex, and if the parameter setting is improper, it is easy to deviate from the high-quality solution and fall into a local optimum. The non-dominated genetic algorithm is widely used because of its low computational complexity, fast convergence speed and strong global search ability. However, its single chromosome mutation mechanism leads to its insufficient local search ability and makes it easy for it to fall into a local optimum. In contrast, HBMO has seen wide interest because of its simple parameter setting, fast convergence speed and strong local search ability [34–36]. Therefore, aiming at the practical problems of low efficiency in the process decision-making of aerospace complex components, the difficulty in adapting to changeable requirements, and the easy-to-appear machine tool bottlenecks and resource conflicts, this paper designs an improved bee mating algorithm to solve the model from the perspective of the conflict between process planning and job-shop scheduling integration optimization objectives.

In encoding, a four-layer encoding mode based on a feature machining sequence, feature machining method, process optional machining machine tool and process optional machining tool are adopted, wherein the feature layer adopts a priority-based encoding mode to avoid the problem of infeasible solution caused by a crossover and mutation operation. On this basis, the greedy algorithm is introduced to decode and obtain the completion time of each scheme. The breeding mechanism of five kinds of worker bees of breeding young bees is designed to make the algorithm easily jump out of the local optimum. The queen bee set preservation strategy is set to make the non-inferior solution participate in the generation of the next generation, which effectively ensures the diversity of the population. The whole framework of the improved HBMO algorithm is given in Figure 1.

#### 4.2. Encoding and Decoding Design

To solve the problem of IPPS of complex aerospace components, it is important to establish a certain relationship between practical problems and chromosome gene structure. According to the complexity of the problem, this paper chooses four-layer integer encoding. Each bee in the population represents a scheme, and the chromosome of each bee contains four encoding sequences, which are the feature sequence, processing method sequence, machine tool sequence and tool sequence. The part-related information is given in Table 2.

The machining features of the part, the optional machining methods for each machining feature, the processes included in each machining method, the available machine tools, the tools corresponding to each process, and the machining sequence constraints of machining features are shown in Table 2. According to the part's machining feature constraints in Table 3, the feature machining priority matrix ($G_p$) can be constructed:

$$G_p = \begin{vmatrix} / & 0 & 1 & 0 & 0 & 0 & 0 & 0 & 0 & 0 & 0 \\ 0 & / & 0 & 0 & 0 & 0 & 0 & 0 & 0 & 0 & 0 \\ 0 & 0 & / & 0 & 0 & 0 & 0 & 0 & 0 & 0 & 0 \\ 0 & 0 & 0 & / & 0 & 0 & 0 & 0 & 0 & 0 & 0 \\ 0 & 0 & 0 & 0 & / & 1 & 0 & 0 & 0 & 0 & 0 \\ 0 & 0 & 0 & 0 & 0 & / & 0 & 0 & 0 & 0 & 0 \\ 0 & 0 & 0 & 0 & 1 & 0 & / & 0 & 0 & 0 & 0 \\ 0 & 0 & 0 & 0 & 0 & 0 & 0 & / & 1 & 0 & 0 \\ 0 & 0 & 0 & 0 & 0 & 0 & 0 & 0 & / & 0 & 0 \\ 0 & 0 & 0 & 0 & 0 & 0 & 0 & 0 & 0 & / & 0 \\ 0 & 0 & 0 & 0 & 0 & 0 & 0 & 0 & 0 & 0 & / \end{vmatrix}$$

$$\text{Elements in a matrix, } g_{ij} = \begin{cases} 1 & \text{feature } F_i \text{ must be processed before feature } F_j \\ 0 & \text{other} \end{cases} \quad i \neq j.$$

**Table 2.** Part processing information.

| Parts | Machining Characteristics | Optional Machining Method | Operation Corresponding to Optional Method | Optional Machine Tool | Corresponding Processing Time of Machine Tool | Optional Cutter | Constraint Relation |
|---|---|---|---|---|---|---|---|
| Part 1 | F1; | Meth1 | 1op1 | M1, m2 | 13, 4 | T1, t2, t3 | Before F3 |
| | | Meth2 | 1op2 | M2, m3 | 4, 3 | T1, t2 | |
| | | | 1op3 | M1, m2 | 5, 4 | T3 | |
| | F2; | Meth1 | 1op4 | M4, m7, m8 | 7, 6, 4 | T7, t8, t9 | |
| | | Meth2 | 1op5 | M4, m5 | 3, 2 | T7 | |
| | | | 1op6 | M7, m8 | 3, 4 | T3, t4 | |
| | About F3 | Meth1 | 1op7 | M7, m8 | 6, 4 | T4, t5, t6 | |
| Part 2 | No. F4 | Meth1 | 2op1 | M2, m10 | 10, 5 | T3, t15, t16 | Before F6 |
| | | Meth2 | 2op2 | M8, m9 | 4, 6 | T19, t20 | |
| | No. F5 | Meth1 | 2op3 | M3, m5 | 3, 7 | T2, t7, t13 | |
| | Federal 6 | Meth1 | 2op4 | M1, m7 | 3, 9 | T5, t7 | |
| | | | 2op5 | M5, m10 | 6, 8 | T3, t8 | |
| | Federal 7 | Meth1 | 2op6 | M3, m6 | 4, 8 | T2, t5 | Before F5 |
| Part 3 | No. F8 | Meth1 | 3op1 | M6, m10 | 4, 5 | T11, t15, t16 | Before F9 |
| | | Meth2 | 3op2 | M8, m9 | 4, 6 | T19, t20 | |
| | No. F9 | Meth1 | 3op3 | M4, m9 | 3, 4 | T6, t7, t12 | |
| | No. F10 | Meth1 | 3op4 | M1, m3 | 3, 4 | T5, t6 | |
| | | | 3op5 | M2, m6 | 6, 4 | T4, t6 | |
| | No. F11 | Meth1 | 3op6 | M2, m3 | 2, 4 | T6, t7, t12 | |

**Table 3.** Related parameters and definitions in the model.

| Symbol | Meaning | Symbol | Meaning |
|---|---|---|---|
| $n_p$ | Number of parts to be processed | $op_{ijkh}$ | The first working procedure of the first machining method of the first characteristic unit of the part, $P_i\, j\, k\, h\ h = 1, 2, \ldots, op\_num_{ijk}$ |

**Table 3.** *Cont.*

| Symbol | Meaning | Symbol | Meaning |
|---|---|---|---|
| $P_i$ | The first part in the parts to be processed, $i$; $i = 1, 2, \ldots, n_p$ | $TimeM_{ijkhu}$ | The machining time on the machine tool of the first working procedure of the first machining method of the first feature unit of the part, $P_i$ $j$ $k$ $h$ $M_u$ |
| $n_m$ | Number of machine tools available for machining | $CostM_u$ | Cost per unit time of machine tool ($M_u$) |
| $n_t$ | Number of tools available for machining | $CostT_v$ | Cost per unit time of tool use ($T_v$) |
| $M_u$ | The first machine tool, $u$; $u = 1, 2, \ldots, n_m$ | $C_{ijkhu}$ | The earliest completion time of the process on the machine tool ($op_{ijkh}$ $M_u$) |
| $T_v$ | The first cutter, $v$; $v = 1, 2, \ldots, n_t$ | $S_{ijkhu}$ | Start time of working procedure on machine tool ($op_{ijkh}$ $M_u$) |
| $fea\_num_i$ | The total number of feature units contained in the part $P_i$; $i = 1, 2, \ldots, n_p$ | $X_{ijk}$ | Decision variable; if selected as the machining method, take 1, otherwise take 0 ($fea_{ij}$ $method_{ijk}$) |
| $fea_{ij}$ | The first feature unit of a part, $P_i$; $j$ $i = 1, 2, \ldots, n_p, j = 1, 2, \ldots, fea\_num_i$ | $Y_{ijkhu}$ | Decision variable; if selected as a machine tool, take 1, otherwise take 0 ($op_{ijkh}$ $M_u$) |
| $method\_num_{ij}$ | Number of machining methods for the first feature unit of parts, $P_i$ $j$ | $Z_{ijkhv}$ | Decision variable; if the tool is selected, take 1, otherwise take 0 ($op_{ijkh}$ $T_v$) |
| $op\_num_{ijk}$ | Number of processes included in the first machining method of the first feature unit of the part, $P_i$ $j$ $k$ | $method_{ijk}$ | The first processing method of the first characteristic unit of the part $P_i$ $j$ $k$ $k = 1, 2, \ldots, method\_num_{ij}$ |

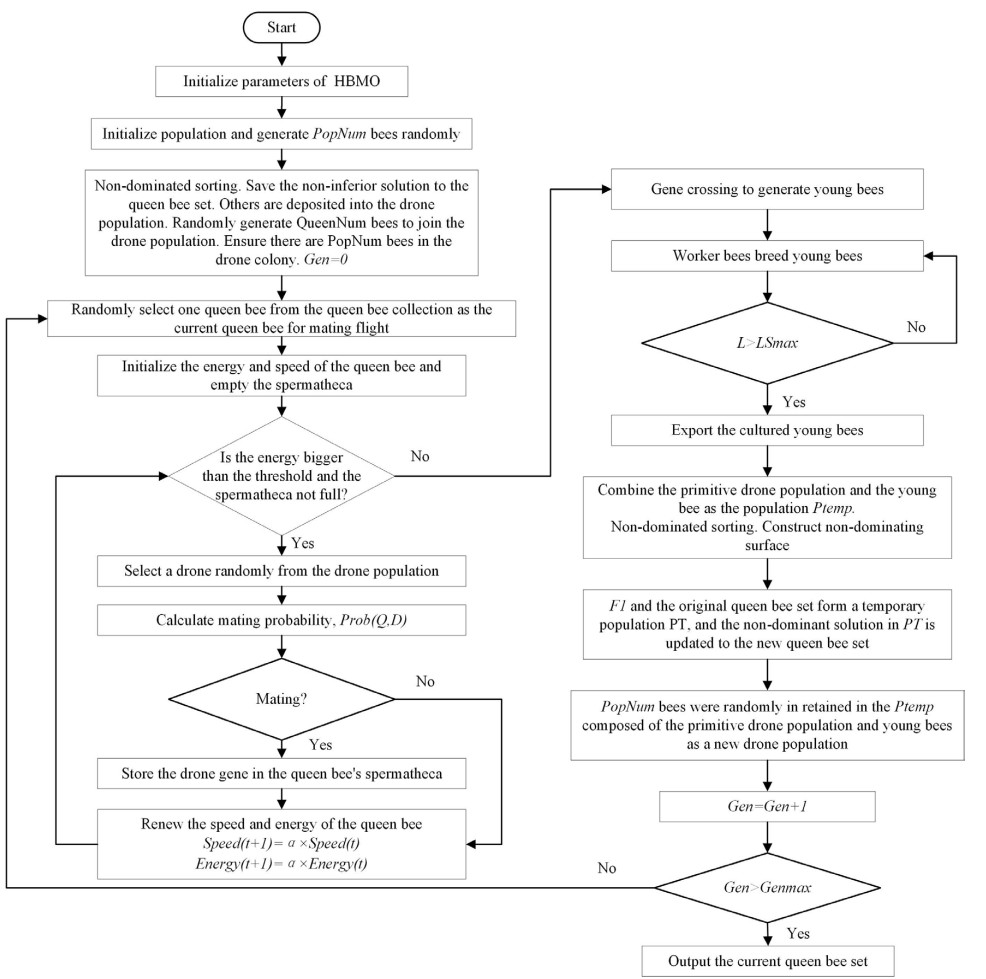

**Figure 1.** HBMO flow chart for solving process reconfiguration model of aerospace complex components.

### 4.3. Encoding Design

#### 4.3.1. Encoding of Feature Layer

The encoding of a feature layer is based on feature priority. The value of the gene locus in a chromosome represents the priority value of the feature of the gene locus, and the length of feature layer encoding is the total number of features of each part. From Table 2, it can be seen that there are 11 features in the three parts, so the length of feature layer encoding is 11, and the value of each gene location is a unique integer between $[1, 11]$, forming an 11-number sequence. The larger the value, the higher the feature priority represented by the corresponding gene location. Figure 2 is a feasible feature encoding sequence.

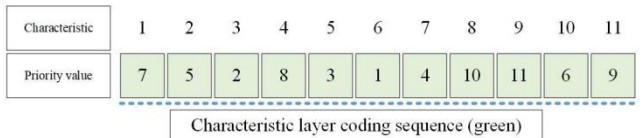

**Figure 2.** Feature layer encoding sequence.

#### 4.3.2. Encoding of Machining Method Layer

The length of the encoding sequence of the machining method layer is the total number of features of each part, and each gene location represents the optional machining method of the feature. If a feature can choose a machining method ($m$), a random integer of $[1, 11]$ is generated at the corresponding gene location of the feature. Figure 3 is a feasible processing method encoding sequence.

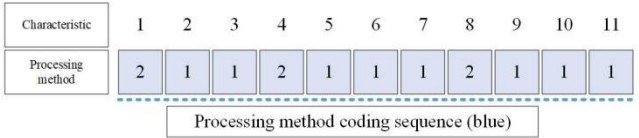

**Figure 3.** Encoding sequence of machining method layer.

#### 4.3.3. Machine Tool Layer Encoding

The length of the machine tool layer encoding sequence is the total number of all machining processes of each part, and each gene position represents the sequence number of machine tools that can be selected in the machining process. If a machine tool's $n$ can be selected in a certain machining process, a random integer of $[1, n]$ is generated on the gene position corresponding to the process. Figure 4 is a feasible machine layer encoding sequence.

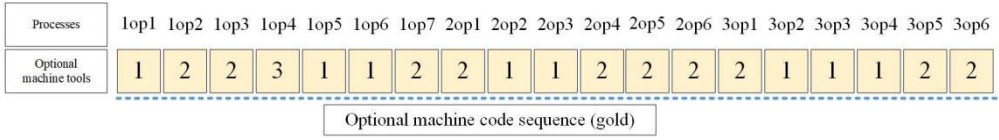

**Figure 4.** Encoding sequence of machine tool layer.

#### 4.3.4. Encoding of Tool Layer

The length of the cutter layer encoding sequence is the total number of all machining processes of each part, and each gene position represents the sequence number of cutters that can be selected in the machining process. If $l$ cutters can be selected in a certain machining process, a random integer of $[1, l]$ is generated on the gene position corresponding to the process. Figure 5 is a feasible tool layer encoding sequence.

Therefore, each bee chromosome should contain the four sequences shown in Figure 6:

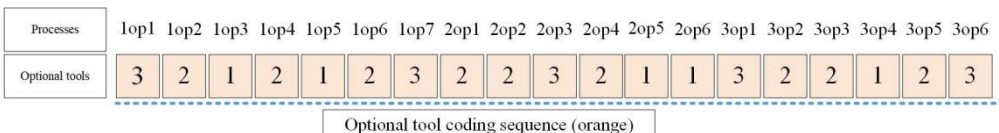

**Figure 5.** Tool layer encoding sequence.

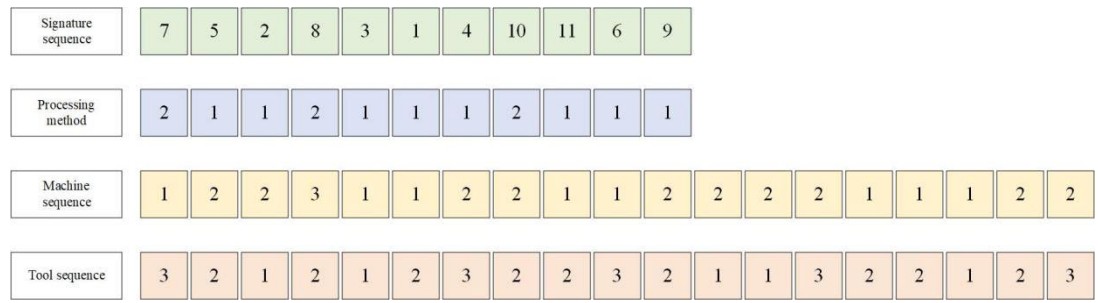

**Figure 6.** Four sequences of honeybee chromosomes.

### 4.4. Decoding Operation

According to the above encoding method, each bee can obtain four encoding sequences, which are decoded by combining the part's processing information and feature-processing priority of matrix $G_p$ in Table 2.

#### 4.4.1. Decoding of Feature Layer

Step 1: Find out the priority value of feature $F_i, \ldots, F_j$ in the encoding sequence corresponding to column $i, \ldots, j$ with all elements of the $G_p$ matrix being 0.

Step 2: Compare the priority values of $F_i, \ldots, F_j$ in the sequence, output the feature corresponding $F_k (k \in i, \ldots, j)$ to the maximum value, and then set the priority value of the feature to zero in the encoding sequence.

Step 3: Set row $k$ in matrix $G_p$ to zero.

Step 4: Repeat the above steps until all features are output.

According to the encoding sequence of the above feature layer, the decoded processing feature sequence should be:

$$F_8 \rightarrow F_9 \rightarrow F_{11} \rightarrow F_4 \rightarrow F_1 \rightarrow F_{10} \rightarrow F_2 \rightarrow F_7 \rightarrow F_5 \rightarrow F_3 \rightarrow F_6$$

#### 4.4.2. Decoding of Machining Method Layer

According to the number on the gene position, the machining method of the feature in the part's machining information table is searched. If the first digit in the sequence is two, it means that feature $F_1$ selects the second machining method, and if the second digit is one, it means that feature $F_2$ selects the first machining method and outputs the process corresponding to the machining method. According to the encoding and decoding of the above processing method layer, it should be:

$$F_1(1op2 \rightarrow 1op3) - F_2(1op4) - F_3(1op7) - F_4(2op2) - F_5(2op3) - F_6(2op4 \rightarrow 2op5) - F_7(2op6)$$
$$- F_8(3op2) - F_9(3op3) - F_{10}(3op4 \rightarrow 3op5) - F_{11}(3op6)$$

#### 4.4.3. Decoding of Machine Tool Layer

According to the number on the gene position, the processing machine tool of the corresponding process in the part's processing information table is searched. If the fourth digit in the sequence is three, it means that process $1op4$ selects the third machine tool ($m8$) in the optional machine tool number. According to the encoding of the above machine tool layer, after decoding, it is:

$1op1(m1) - 1op2(m3) - 1op3(m2) - 1op4(m8) - 1op5(m4) - 1op6(m7) - 1op7(m8) - 2op1(m10)$
$- 2op2(m8) - 2op3(m3) - 2op4(m7) - 2op5(m10) - 2op6(m6) - 3op1(m10) - 3op2(m8) - 3op3(m4)$
$- 3op4(m1) - 3op5(m6) - 3op6(m3)$

### 4.4.4. Decoding of Tool Layer

According to the number on the gene position, the machining of the corresponding process in the part's machining information table is searched. If the seventh digit in the sequence is three, it means that process $1op7$ selects the third tool ($t6$) in the optional tool number. According to the encoding of the above tool layer, the decoded one is:

$1op1(t3) - 1op2(t2) - 1op3(t3) - 1op4(t8) - 1op5(t7) - 1op6(t4) - 1op7(t6) - 2op1(t15) - 2op2(t20)$
$- 2op3(t13) - 2op4(t7) - 2op5(t3) - 2op6(t2) - 3op1(t16) - 3op2(t20) - 3op3(t7) - 3op4(t5)$
$- 3op5(t6) - 3op6(t12)$

### 4.4.5. Final Decoding

According to the decoding of the above four sequences, the feature sequence is decoded first to obtain the sequence of the processing features. Then, the processing method sequence is decoded to obtain the processing procedure of each feature, and the sequences of processing features are combined to obtain the process sequence. Finally, the machine tool sequence and tool sequence are decoded to obtain the machine tool and tool used in each process. The final process route can be obtained as follows:

$(3op2, m8, t20) \rightarrow (3op3, m4, t7) \rightarrow (3op6, m3, t12) \rightarrow (2op2, m8, t20) \rightarrow (1op2, m3, t2) \rightarrow (1op3, m2, t3)$
$\rightarrow (3op4, m1, t5) \rightarrow (3op5, m6, t6) \rightarrow (1op4, m8, t8) \rightarrow (2op6, m6, t2) \rightarrow (2op3, m3, t13) \rightarrow (1op7, m8, t6)$
$\rightarrow (2op4, m7, t7) \rightarrow (2op5, m10, t3)$

The first digit in brackets is the machining process, the second digit is the machine tool used in this process, and the third digit is the tool used in this process.

### 4.4.6. Chromosome Decoding Based on Greedy Algorithm

The above decoding method can only obtain semi-active scheduling, which is not the optimal condition of active scheduling. Therefore, the greedy algorithm is used for decoding. First, the following symbols are defined: $M$ indicates the total number of machine tools; $Op_{ij}$ indicates the $j$ process of the first $i$ part, $aS_{ij}$ indicates the feasible start time of the operation $Op_{ij}$, $S_{ij}$ is the specific start time of the operation $Op_{ij}$, $u$ is the processing machine of process $Op_{ij}$, $T_{iju}$ is the processing time of process $Op_{ij}$ on the machine's $u$, and $C_{ij}$ is the completion time of the operation $Op_{ij}$, $C_{ij} = S_{ij} + T_{iju}$. The specific algorithm is as follows:

Step 1: According to the above decoded process sequence and the corresponding machine tool and processing time, determine the processing machine tool set for each part and the processing process set for each machine tool.

Step 2: Calculate the theoretical start time ($aS_{ij}$) of each process $Op_{ij}$ and the completion time of the process in the part $i$ before $Op_{ij}$, $aS_{ij} = C_{i(j-1)}$.

Step 3: Check the idle time of the process $Op_{ij}$ on the processing machine and obtain a series of idle time regions, $[t_s, t_e]$. If $\max(aS_{ij}, t_s) + T_{iju} \leq t_e$, then make $S_{ij} = \max(aS_{ij}, t_s)$; otherwise, check the next area. If all areas are not satisfied, then $S_{ij} = \max\{aS_{ij}, C(Op_{ij} - 1)\}$, in which $C(Op_{ij} - 1)$ is the completion time of the previous process on the same machine as $Op_{ij}$.

Step 4: From this, the start time ($S_{ij}$) and completion time ($C_{ij} = S_{ij} + T_{iju}$) of each operation can be obtained.

### 4.5. Young Bee Formation Stage

After the queen bee's mating flight, the genotypes of different drones are preserved in the fertilization sac of the queen bee, and it is necessary to crossoperate the genes to

produce new young bees. Each bee contains four sequences, which need to be crossed separately. In these four sequences, the machining method layer, machine tool layer and tool layer have the same characteristics and can adopt the same intersection mode, so the intersection of feature layer and machining method layer is described in detail.

### 4.6. Crossover Operation of Feature Layer

The crossover operation of the feature layer is revealed by Figure 7. Two crosspoints (red dots in Figure 7) are selected, and the characteristic sequence of the queen bee is divided into front, middle and back segments. The middle genes of the queen bee (the values between crosspoints) are directly copied to the corresponding middle gene loci of young bees, and the front and back genes of queen bee are copied to the front and back gene loci of young bees according to the sequence of male peak sequences.

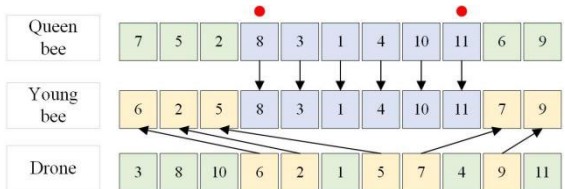

**Figure 7.** Intersection of feature layers.

### 4.7. Crossing of Process Layers

The crossover operation of the machining method layer is shown in Figure 8. Two crosspoints (red dots in Figure 8) are selected, and the processing sequence of the queen bee and male peak is divided into front, middle and back segments. The middle gene of the queen bee (i.e., the value between crosspoints) is directly copied to the middle gene locus corresponding to young bees, and the front and back genes of male bees are directly copied to the front and back gene locus corresponding to young bees.

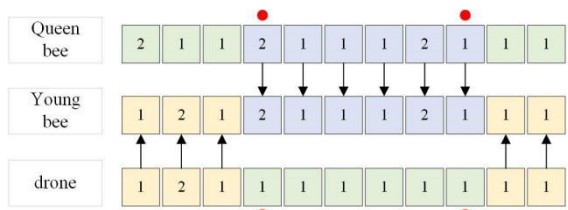

**Figure 8.** Intersection of machining method layers.

The crossover operation of the machine tool layer and the tool layer is similar to the crossover operation of the machining method layer and will not be repeated here.

### 4.8. Cultivation Stage of Worker Bees

In the improved HBMO algorithm, each worker bee is equivalent to a local search strategy. When the young bees are produced, it is necessary to cultivate each young bee. Based on the design of the encoding mode, the exchange operation ($V1$), insertion operation ($V2$) and mutation operation ($V3$) are designed.

(1) Exchange operation, $V1$: randomly select two different positions on the feature encoding sequence in young bees, and exchange the values at these two positions, as shown in Figure 9.

(2) Insertion operation, $V2$: Randomly select a position on the feature encoding sequence in young bees, and insert the value at this position into any position on the sequence, and postpone the value of the gene site after insertion. The operation process is listed in Figure 10.

(3) Mutation operation, $V3$: A gene locus in the processing method layer is randomly selected, and another processing method is selected according to its optional processing method. In the same way, one gene locus in the machine tool layer and tool layer is randomly selected, and another processing resource is selected according to its optional processing resources. The mutation operation process is given in Figure 11.

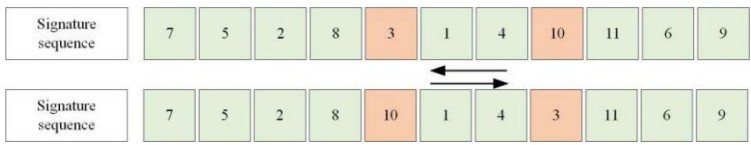

**Figure 9.** Switch operation, $V1$.

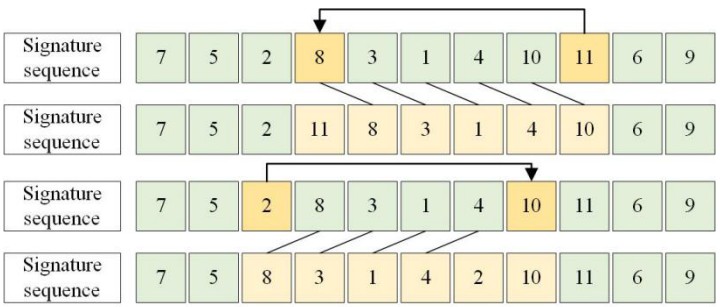

**Figure 10.** Insert Operation, $V2$.

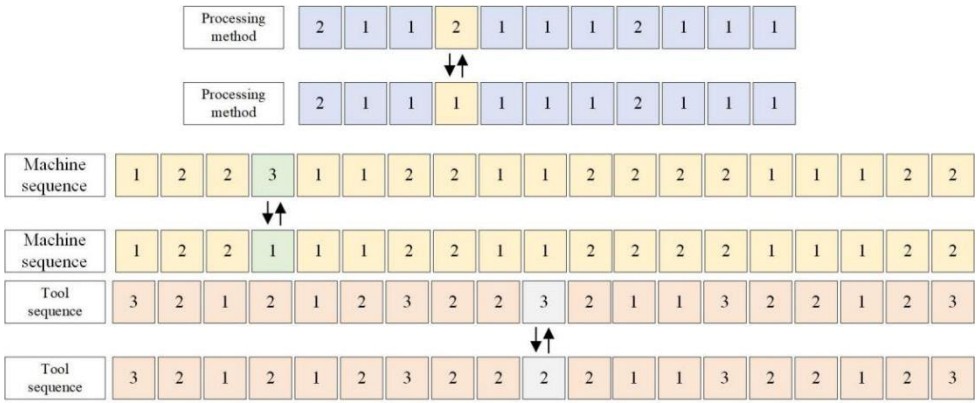

**Figure 11.** Mutation operation, $V3$.

The above operations can be carried out alone or in combination, i.e., $V1$, $V2$, $V3$, $V1 + V3$, $V2 + V3$, so the number of worker bees is five; $W_i = \{W_1, W_2, W_3, W_4, W_5\}$.

Based on the above breeding strategy, the detailed steps for worker bees to breed young bees are as follows:

Step 1: Set the maximum number of iterations for worker bees to breed young bees, so that $LS_{\max}$, $t = 1$.

Step 2: Randomly select a worker bee from an optional worker bee set, in which the young bee $\theta$ is cultivated to obtain a new young bee $\theta'$; $W_i$ $i \in \{1, 2, 3, 4, 5\}$ $W_i$.

Step 3: If the new young bee $\theta'$ can dominate the young bee $\theta$, use the young bee $\theta'$ instead of the young bee $\theta$, otherwise keep the young bee $\theta$.

Step 4: If not, jump to Step 2. Otherwise, the breeding process is terminated and the current young bees are exported; $t = t + 1$ $t < LS_{\max}$.

## 5. Case Analysis

### 5.1. Process Parameters

An aerospace complex component manufacturer mainly produces complex components such as a fuze seat shell, hydraulic servomechanism channel body, hydraulic valve

oil cylinder body, etc. The sample diagrams of the three parts are shown in Figure 12. Its production workshop is mainly composed of 12 pieces of machining equipment, including three CNC lathes, five milling machines, two boring machines and two vertical machining centers. At present, it is necessary to arrange the process route of six kinds of aerospace complex components, and the processing methods, available machine tools and available cutting tools of each part are shown in Table 4. The unit time use cost of each machine and tool is shown in Tables 5 and 6.

**Table 4.** Parts' Processing Technology Information.

| Parts | Feature Number | Optional Machining Method | Process Corresponding to Machining Method | Optional Machine Tool | Machining Time Corresponding to Machine Tool (h) | Optional Cutter | Order Constraints between Features |
|---|---|---|---|---|---|---|---|
| Part 1 | F1 | Meth1 | 1op1 | 1/2/3 | 0.6/0.8/0.8 | T1/T2/T3 | Before all the features |
| | | | 1op2 | 11/12 | 0.7/0.7 | T24/T25 | |
| | | Meth2 | 1op3 | 2/3 | 1.1/1.3 | T3/T4 | |
| | F2 | Meth1 | 1op4 | 11/12 | 0.8/0.8 | T24/T26 | Before F3/F4/F6 |
| | F3 | Meth1 | 1op5 | 1/2/3 | 0.5/0.6/0.6 | T2/T4 | |
| | F4 | Meth1 | 1op6 | 1/2/3 | 0.6/0.6/0.6 | T2/T4/T5 | |
| | | Meth2 | 1op7 | 9/10 | 0.6/0.6 | T15/T16/T17 | |
| | | | 1op8 | 9/10 | 0.7/0.7 | T18/T19 | |
| | F5 | Meth1 | 1op9 | 1/2/3 | 0.9/8/0.9 | T1/T3/T4 | |
| | | Meth2 | 1op10 | 4/5 | 0.4/0.6 | T9/T11/T12/T13 | |
| | F6 | Meth1 | 1op11 | 1/2/3 | 0.7/0.5/0.9 | T1/T5 | |
| | | Meth2 | 1op12 | 6/7/8 | 0.4/0.5/0.5 | No. 6/T7/T12/T14 | |
| Part 2 | F7 | Meth1 | 2op1 | 1/2/3 | 0.5/0.3/0.6 | T2/T3/T4/T5 | Before all the features |
| | | | 2op2 | 11/12 | 0.6/0.6 | T25/T26 | |
| | F8 | Meth1 | 2op3 | 4/5/6 | 0.8/0.5/0.9 | No. 7/T8/T10/T11/T12 | Before F9 |
| | | | 2op4 | 6/7/8 | 0.7/0.6/0.6 | No. 6/T8/T10/T12 | |
| | F9 | Meth1 | 2op5 | 4/5 | 0.9/8.8 | T9/T10/T14 | Before F10/F11/F12 |
| | | | 2op6 | 4/5/6 | 0.8/0.6/0.7 | T7/T11/T12/T13 | |
| | | Meth2 | 2op7 | 5/6 | 0.7/0.7 | T9/T12/T13/T14 | |
| | | | 2op8 | 5/6 | 0.7/0.7 | T9/T12/T13/T14 | |
| | F10 | Meth1 | 2op9 | 1/2 | 0.6/0.6 | No. T9/T10/T11 | |
| | | Meth2 | 2op10 | 1/2/3 | 0.6/0.6/0.5 | T4/T5 | |
| | | | 2op11 | 11/12 | 0.9/9 | T2/T3/T4/T5 | |
| | F11 | Meth1 | 2op12 | 9/10 | 0.6/0.6 | T24/T25/T26 | |
| | | Meth2 | 2op13 | 9/10 | 0.8/0.8 | T20/T21 | |
| | | | 2op14 | 9/10 | 0.6/0.6 | T18/T20/T21 | |
| | F12 | Meth1 | 2op15 | 1/2/3 | 0.8/0.6/0.7 | T1/T4/T5 | |

**Table 4.** *Cont.*

| Parts | Feature Number | Optional Machining Method | Process Corresponding to Machining Method | Optional Machine Tool | Machining Time Corresponding to Machine Tool (h) | Optional Cutter | Order Constraints between Features |
|---|---|---|---|---|---|---|---|
| Part 3 | F13 | Meth1 | 3op1 | 4/5 | 0.8/0.6 | No. 7/T8/T11/T14 | |
| | | Meth2 | 3op2 | 9/10 | 0.7/0.7 | T15/T16/T17 | |
| | | | 3op3 | 9/10 | 0.9/9 | T22/T23 | |
| | F14 | Meth1 | 3op4 | 6/7/8 | 0.6/0.7/0.7 | No. 7/T8/T9/T10 | Before all the features |
| | | | 3op5 | 11/12 | 0.3/0.3 | T24/T25 | |
| | | Meth2 | 3op6 | 1/2/3 | 0.6/0.8/0.8 | T1/T2/T4/T5 | |
| | F15 | Meth1 | 3op7 | 4/5 | 0.5/0.5 | T10/T11/T12 | Before F17 |
| | | Meth2 | 3op8 | 9/10 | 0.3/0.3 | T18/T19/T20/T21 | |
| | F16 | Meth1 | 3op9 | 1/2/3 | 0.7/0.8/0.6 | T1/T2/T3 | |
| | F17 | Meth1 | 3op10 | 1/2/3 | 0.8/0.6/0.6 | T2/T3/T4 | |
| Part 4 | F18 | Meth1 | 4op1 | 6/7/8 | 0.4/0.4/0.5 | T6/T7/T8 | Before all the features |
| | | | 4op2 | 11/12 | 0.6/0.6 | T25/T26 | |
| | F19 | Meth1 | 4op3 | 4/5/7/8 | 0.5/0.5/0.6/0.6 | No. 7/T8/T9/T10/T13 | |
| | F20 | Meth1 | 4op4 | 4/5 | 0.6/0.5 | T10/T11/T14 | |
| | | Meth2 | 4op5 | 9/10 | 0.3/0.3 | T15/T16/T18/T19 | |
| | F21 | Meth1 | 4op6 | 9/10 | 0.6/0.7/0.7 | 6/T7/T9/T11/T12 | |
| | | Meth2 | 4op7 | 1/2/3 | 0.6/0.5 | T15/T16/T17 | |
| | | | 4op8 | 4/5/6 | 0.6/0.8 | T18/T20/T21 | |
| | F22 | Meth1 | 4op9 | 1/2/3 | 0.4/0.4/0.4 | T2/T3/T4 | |
| | | Meth2 | 4op10 | 4/5/6 | 0.5/0.5/0.6 | T7/T8/T11/T12/T14 | |
| | | | 4op11 | 5/6/7 | 0.6/0.4/0.4 | No. 7/T8/T12/T13 | |
| Part 5 | F23 | Meth1 | 5op1 | 1/2/3 | 0.5/0.6/0.6 | T1/T2/T4/T5 | Before all the features |
| | | | 5op2 | 1/2/3 | 0.5/0.5/0.5 | T2/T3/T5 | |
| | | Meth2 | 5op3 | 4/5 | 0.4/0.4 | T9/T10/T11/T12 | |
| | F24 | Meth1 | 5op4 | 6/7/8 | 0.6/0.7/0.6 | T6/T7/T8 | |
| | F25 | Meth1 | 5op5 | 1/2/3 | 0.6/0.6/0.6 | T2/T3 | |
| | | | 5op6 | 4/5 | 0.3/0.4 | T10/T12/T13/T14 | |
| | F26 | Meth1 | 5op7 | 6/7/8 | 0.4/0.6/0.7 | No. 6/T8/T9/T11 | |
| | F27 | Meth1 | 5op8 | 2/3 | 0.5/0.5 | T3/T4/T5 | Before F24 |
| | | | 5op9 | 6/7/8 | 0.6/0.4/0.5 | No. 7/T8/T11/T14 | |
| | | Meth2 | 5op10 | 11/12 | 0.5/0.5 | T24/T25/T26 | |
| | F28 | Meth1 | 5op11 | 1/2/3 | 0.6/0.5/0.5 | T2/T4/T5 | |

**Table 4.** *Cont*.

| Parts | Feature Number | Optional Machining Method | Process Corresponding to Machining Method | Optional Machine Tool | Machining Time Corresponding to Machine Tool (h) | Optional Cutter | Order Constraints between Features |
|---|---|---|---|---|---|---|---|
| Part 6 | F29 | Meth1 | 6op1 | 5/8 | 1.0/0.8 | T4/T5/T7/T8 | Before F33 |
| | F30 | Meth1 | 6op2 | 9/10 | 0.6/0.4 | T15/T16/T17 | Before F33 |
| | | | 6op3 | 4/6 | 0.3/0.5 | T7/T8 | |
| | | Meth2 | 6op4 | 9/10 | 0.6/0.4 | T15/T16/T17 | |
| | | | 6op5 | 6/7/8 | 0.5/0.5/0.4 | T6/T7/T8 | |
| | F31 | Meth1 | 6op6 | 4/7/9/10 | 0.6/0.7/0.8/0.8 | T12/T13/T14 | Before all the features |
| | | Meth2 | 6op7 | 4/5/7 | 0.6/0.5/0.7 | T12/T13/T14 | |
| | | Meth3 | 6op8 | 1/2/3 | 0.6/0.6/0.8 | T12/T13/T14 | |
| | F32 | Meth1 | 6op9 | 4/5/6 | 0.8/0.9/0.7 | No. T9/T12/T14 | Before F29 |
| | | Meth2 | 6op10 | 9/10 | 0.4/0.4 | T18/T19 | |
| | | | 6op11 | 7/8 | 0.6/0.4 | T6/T7/T8 | |
| | F33 | Meth1 | 6op12 | 1/2/3 | 0.8/0.6/0.6 | T1/T2/T3/T4/T5 | |

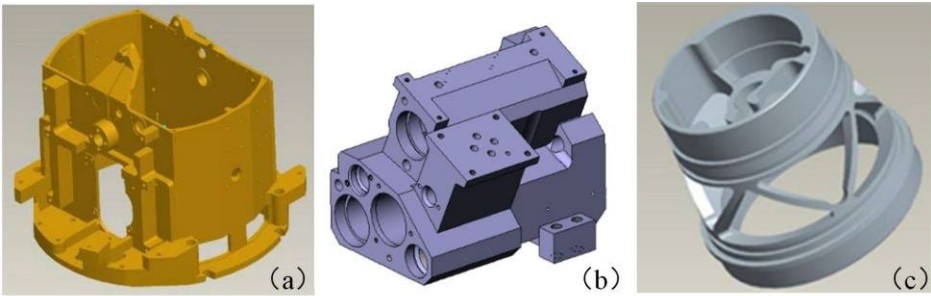

**Figure 12.** Sample parts. (**a**) Shell; (**b**) Passage Body; (**c**) Fuze Seat.

**Table 5.** Processing equipment information and cost per unit time.

| Machine Tool Number | Name of Machine Tool | Type of Machine Tool | Machine Tool Cost per Unit Time (h/CNY) |
|---|---|---|---|
| China Net | Numerical control lathe | CK6163 | 12 |
| China Net | Numerical control lathe | CAK4085DI | 16 |
| China Net | Horizontal CNC lathe | CTX310V1 | 16 |
| China Net | Ordinary milling machine | 3 | 14 |
| Grand game | Three-axis NC vertical milling | KVC1050MA | 10 |
| China Net | Vertical CNC milling machine | DMU 35M | 16 |
| China Net | Drilling and milling machining center | TLV-500 | 12 |
| China Net | Numerical control vertical milling machine | VMP-32A | 12 |
| China Net | Boring and milling machining center | 1150 | 12 |
| China Net | Five-axis boring and milling machining center | UCP800 | 12 |
| Grand game | Vertical machining center | DMC63V | 16 |
| China Net | NC machining center | GX1000 Plus | 16 |

**Table 6.** Tool cost per unit time.

| Tool Number | Tool Cost per Unit Time (h/CNY) | Tool Number | Tool Cost per Unit Time (h/CNY) |
|---|---|---|---|
| T1 | 5 | T14 | 6 |
| T2 | 4 | T15 | 5 |
| T3 | 5 | T16 | 4 |
| T4 | 4 | T17 | 5 |
| T5 | 9 | T18 | 7 |
| T6 | 8 | T19 | 9 |
| T7 | 9 | T20 | 8 |
| T8 | 4 | T21 | 6 |
| T9 | 5 | T22 | 6 |
| T10 | 4 | T23 | 4 |
| T11 | 4 | T24 | 5 |
| T12 | 5 | T25 | 7 |
| T13 | 6 | T26 | 4 |

*5.2. Efficiency Analysis of Improved HBMO Algorithm*

To minimize the makespan, the machining time and the machining cost, we solve the above cases by using the HBMO algorithm. Based on Matlab programming, the IPPS of aerospace complex components in the above example is implemented on a computer with 1.6 G CPU and 8G memory. The main parameters of the algorithm are determined by sensitivity analysis. The level of these major parameters are as follows: $Gen_{max}$ = {150, 200, 250}, $\alpha$ = {0.5, 0.7, 0.9}, *threshold* = {0.001, 0.045, 0.1} and $SperNum$ = {75, 100, 125}. The HBMO algorithm is run 10 times for each parameter combination. The max spread (MS) metric is applied to assess the performance of each combination. The main effect plot is available in Figure 13, and a larger value of MS indicates better performance. After several runs, the following parameters were adopted: iteration times or $Gen_{max}$ = 200 of the algorithm, colony size or $BeeNum$ = 200, $QueenNum$ = 50, energy and speed attenuation coefficient of the queen or $\alpha$ = 0.9, energy threshold or *threshold* = 0.001, seminal vesicle capacity or $SperNum$ = 100, number of young bees or $BroodNum$ = 100, number of worker bees or $WorkerNum$ = 5, and iteration times of worker bees breeding young bees or $LS_{max}$ = 20. The record Pareto solution is set after the program runs independently many times. On this basis, to verify the effectiveness of the improved HBMO algorithm proposed in this paper, we use NSGA-II to solve IPPS, and the Patreo solutions between the HBMO and NSGA-II optimizer in small-scale instances are summarized in Table 7. The parameters of NSGA-II are as follows: the population size is 200, the iteration times are 100, the crossover probability is 0.8, and the mutation probability is 0.1. The Pareto solution front is shown in Figure 14.

From Figure 14, it can be seen that the improved HBMO algorithm retains 10 non-dominated solutions and the NSGA-II algorithm retains 8 non-dominated solutions under the condition of satisfying the number of iterations. It can be seen from Figure 14 that the frontier point of the improved HBMO algorithm is obviously better than that of the NSGA-II algorithm in terms of the makespan, the machining time and the machining cost. Comparing the data in Table 7, it can be seen that the solution obtained by the improved HBMO algorithm is better than that obtained by the NSGA-II algorithm in terms of the makespan and the machining cost, in which the completion time is reduced by 8.47%, and the machining cost is reduced by 2.19%, while the solutions obtained by the two algorithms are similar in terms of the total machine tool running time. The reasons for the above phenomenon are as follows:

(1) The improved HBMO algorithm has set up a queen bee collection preservation mechanism based on the crowding degree in each generation of the iterative process, which can save the current non-inferior solution of the bee colony and participate in the generation of the next generation, ensure the transmission of excellent genes and promote the optimization of the algorithm.

(2) Compared with the single mutation mechanism of the NSGA-II algorithm, the improved HBMO algorithm designs five kinds of breeding mechanisms for young bees in a local search, and breeds each young bee many times, adding better young bees to the population to replace the poor drones, ensuring that the population develops in a better direction after each iteration, which makes the algorithm have a better convergence effect.

Therefore, for solving the IPPS of aerospace complex components, the improved HBMO algorithm can obtain higher-quality solutions, thus effectively shortening the makespan and the machining time of parts and reducing machining cost.

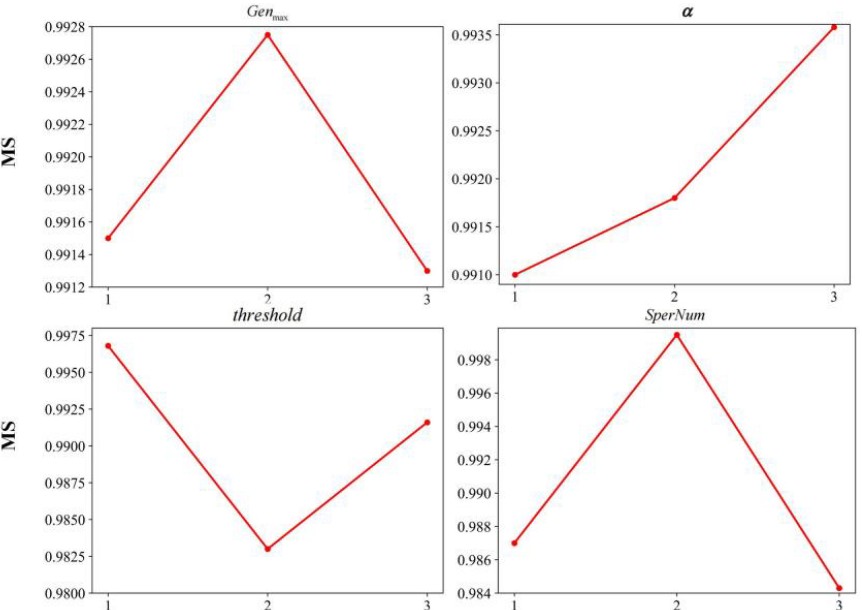

**Figure 13.** Main effect plot of major parameters.

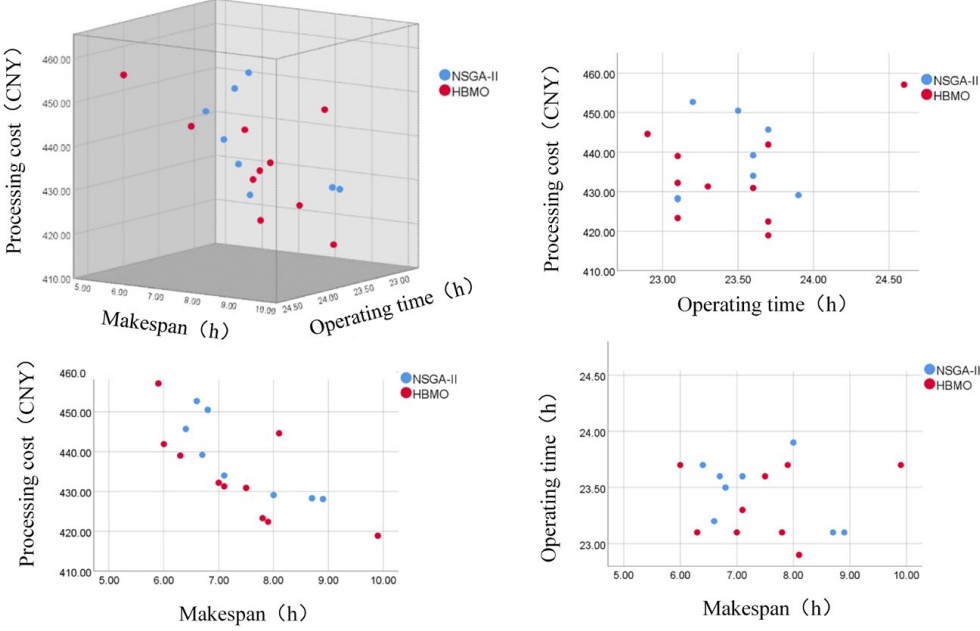

**Figure 14.** Comparison of Pareto frontier points between HBMO and NSGA-II algorithms.

**Table 7.** Comparison of Pareto solutions between improved HBMO algorithm and NSGA-II algorithm.

| Algorithm | F1 (h) | F2 (h) | F3 (CNY) | Algorithm | F1 (h) | F2 (h) | F3 (CNY) |
|---|---|---|---|---|---|---|---|
| HBMO | 7.9 | 23.7 | 422.4 | NSGA II | 8.7 | 23.1 | 428.3 |
| | 6.0 | 23.7 | 441.9 | | 8.0 | 23.9 | 429.1 |
| | 8.1 | **22.9** | 444.6 | | 6.7 | 23.6 | 439.2 |
| | 7.1 | 23.3 | 431.3 | | 7.1 | 23.6 | 434.0 |
| | 6.3 | 23.1 | 439.0 | | **6.4** | 23.7 | 445.7 |
| | 7.5 | 23.6 | 430.9 | | 8.9 | **23.1** | **428.1** |
| | 9.9 | 23.7 | **418.9** | | 6.6 | 23.2 | 452.7 |
| | 7.0 | 23.1 | 432.2 | | 6.8 | 23.5 | 450.5 |
| | **5.9** | 24.6 | 457.1 | | | | |
| | 7.8 | 23.1 | 423.3 | | | | |
| Optimum value | 5.9 | 22.9 | 418.9 | Optimum value | 6.4 | 23.1 | 428.1 |
| Average | 7.35 | 23.48 | 434.16 | Average | 7.4 | 23.46 | 438.45 |

### 5.3. Analysis of Optimization Results

According to the part information, the improved HBMO algorithm is used to solve IPPS. The solution results are shown in Table 8. In order to clearly and intuitively display the distribution of Pareto solutions, a three-dimensional scatter diagram is constructed, as shown in Figure 15.

**Table 8.** Evaluation of process reconfiguration scheme based on TOPSIS evaluation method.

| Serial Number | Maximum Completion Time (h) | Total Running Time of Machine Tool (h) | Cost of Machine Tools and Cutting Tools (CNY) | $D_i^+$ | $D_i^-$ | $C_i$ |
|---|---|---|---|---|---|---|
| 1 | 7.9 | 23.7 | 422.4 | 0.004696 | 0.001815 | 0.278815 |
| **2** | **6.0** | **23.7** | **441.9** | **0.000114** | **0.010967** | **0.989697** |
| 3 | 8.1 | 22.9 | 444.6 | 0.005433 | 0.001439 | 0.209443 |
| 4 | 7.1 | 23.3 | 431.3 | 0.002103 | 0.004164 | 0.664419 |
| 5 | 6.3 | 23.1 | 439.0 | 0.000340 | 0.008575 | 0.961828 |
| 6 | 7.5 | 23.6 | 430.9 | 0.003352 | 0.002759 | 0.451465 |
| 7 | 9.9 | 23.7 | 418.9 | 0.011901 | 0.000189 | 0.015643 |
| 8 | 7.0 | 23.1 | 432.2 | 0.001817 | 0.004605 | 0.717034 |
| 9 | 5.9 | 24.6 | 457.1 | 0.000300 | 0.011865 | 0.975324 |
| 10 | 7.8 | 23.1 | 423.3 | 0.004317 | 0.002101 | 0.327352 |

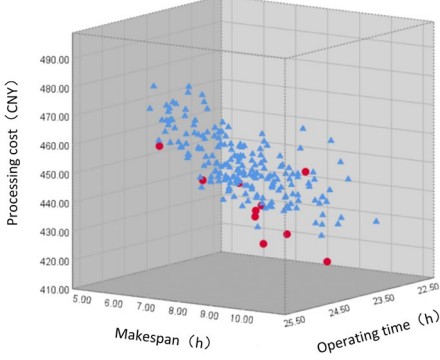

**Figure 15.** Pareto scatter plot of improved HBMO algorithm.

Based on the TOPSIS evaluation method, the Pareto solution set is evaluated, and the weights of the makespan, the machining time and the machining cost are set as $\omega =$

$(0.5, 0.3, 0.2)$. Finally, the approximation degree between each evaluation object and the best scheme and the worst scheme and the approximation degree between each evaluation object ($D_i^+$ and $D_i^-$) and the best scheme's ($C_i$) are shown in Table 9.

**Table 9.** Process route arrangement based on Scheme 2.

| Machine Tool | Processing Sequence |
|---|---|
| China Net | $3op6(T4) \rightarrow 1op5(T2)$ |
| China Net | $2op1(T3) \rightarrow 5op5(T2) \rightarrow 2op9(T4) \rightarrow 5op11(T4) \rightarrow 6op12(T4) \rightarrow 2op15(T4)$ |
| China Net | $1op3(T4) \rightarrow 4op9(T3) \rightarrow 3op9(T2) \rightarrow 3op10(T3)$ |
| China Net | $5op3(T11) \rightarrow 6op3(T7) \rightarrow 5op6(T12) \rightarrow 1op10(T11)$ |
| Grand game | $2op3(T10) \rightarrow 3op1(T11) \rightarrow 2op5(T10) \rightarrow 2op6(T12)$ |
| China Net | $5op4(T7)$ |
| China Net | $4op1(T7) \rightarrow 5op7(T9) \rightarrow 4op6(T9) \rightarrow 4op3(T9) \rightarrow 1op12(T12)$ |
| China Net | $2op4(T10) \rightarrow 6op10(T19) \rightarrow 6op1(T7)$ |
| China Net | $6op6(T13) \rightarrow 4op5(T16) \rightarrow 1op7(T16) \rightarrow 1op8(T18)$ |
| China Net | $6op2(T16) \rightarrow 6op10(T19) \rightarrow 3op8(T19) \rightarrow 2op13(T21) \rightarrow 2op14(T23)$ |
| Grand game | $4op2(T26) \rightarrow 5op10(T25)$ |
| China Net | $2op2(T26) \rightarrow 1op4(T24)$ |

As indicated by the results given in Table 8, although the makespan of Scheme 9 is the shortest, the total running time of its machine tools and the machining cost are on the high side, because the processing time of its part's feature selection processing method is short, but the use cost of its corresponding machine tools and tools is high. Therefore, through using the TOPSIS comprehensive evaluation method to evaluate each optimal scheduling scheme, we can see that the closeness degree ($C_i$) of Scheme 2 is closest to one, so it is the optimal scheduling scheme, and the completion time of all parts is 6.0 h, the total running time of machine tools is 23.7 h, and the use cost of machine tools and tools is 441.9 CNY. The Gantt chart of Scenario 2 is provided in Figure 16, and the scheduling details of Scheme 2 are shown in Table 9.

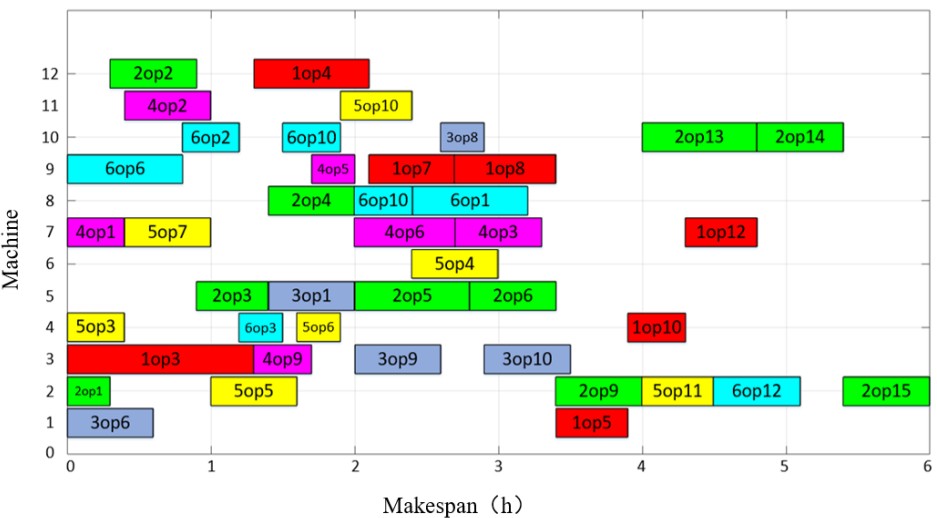

**Figure 16.** Gantt chart of Option 2.

## 6. Conclusions

This work addresses the problem of IPPS for aerospace complex component manufacturing enterprises. A mathematical model is formulated with the optimization objectives of minimizing the makespan, total processing time and machining cost. An improved HBMO algorithm combined with the greedy algorithm is proposed to solve the model. A four-layer encoding and decoding method is designed to improve the adaptability of the algorithm. Moreover, different queen bee preservation strategies and worker bee breed-

ing mechanisms are performed to enhance the search ability of the algorithm as well as avoid a premature crossover and mutation operation. Comparative experiments for the real-world case of the aerospace manufacturing enterprise between the HBMO, NSGA-II and improved HBMO algorithms are conducted to assess the effectiveness of the proposed model and algorithm. Additionally, the results show that the improved HBMO algorithm outperforms its competitors.

It can be seen from the above research that IPPS can effectively help aerospace enterprises to achieve the optimal process decision-making and matching of manufacturing resources rapidly, thus effectively improving overall production capacity. Considering the uncertainties such as emergency orders and machine failures in actual production, the future directions may focus on the IPPS problem in multi-variety and small-batch enterprises ina dynamic environment.

**Author Contributions:** Conceptualization, G.Y., Q.T. and X.J.; methodology, Q.T. and Z.T.; software, Z.T., Q.T. and Y.L.; validation, Q.T. and Z.T.; formal analysis, G.Y. and K.C.; investigation, Q.T. and X.J.; resources, X.J. and W.L.; data curation, G.Y. and Q.T.; writing—original draft preparation, Q.T. and Y.L.; writing—review and editing, Q.T. and X.J.; visualization, Z.T.; supervision, X.J.; project administration, X.J. and P.Y. All authors have read and agreed to the published version of the manuscript.

**Funding:** This research received no external funding.

**Institutional Review Board Statement:** Not applicable.

**Informed Consent Statement:** Not applicable.

**Conflicts of Interest:** The authors declare no conflict of interest.

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
