# Peer review of "Integrated Optimization of Process Planning and Scheduling for Aerospace Complex Component Based on Honey-Bee Mating Algorithm"

_applsci, doi:10.3390/app13085190_

Round 1

Reviewer 1 Report

Integrated Optimization of Process Planning and Scheduling for Aerospace Complex Component Based on Honey-bee Mating algorithm

The Authors propose “an integrated optimization method of complex component process planning and workshop scheduling for aerospace manufacturing enterprises, with the optimization objectives of minimizing the maximum completion time, minimizing the total processing time of machine tools and minimizing the processing cost”. The Authors use “a bee mating optimization algorithm combined with greedy algorithm”.

The paper is well written.

Following, I report the main points:

·       In relation to the Literature described, it is necessary to underline the novelty of the paper; I suggest introducing a comparison table where, by different characteristics, the present work is compared with the Literature.

·       It is not clear how the aerospace market/components production changes the problem analyzed (for example in comparison with the automotive sector). I suggest explaining in the introduction.

·       I suggest introducing in the appendix the sensitivity analysis for the different heuristic parameters.

·       I suggest analyzing the convergence capacity of the bee algorithm with and without the greedy rule (and the related importance of the greedy rule) in comparison with the NSGA-II algoritm.

·       If the problem size is small (only 6 parts), is it possible to generate the optimal solution with an exhaustive method in order to evaluate the convergence to the optimality of the proposed algorithm? 

·       The problem size considered is really very small (6 parts); it is possible to increase the number of parts in order to evaluate the performance of the algorithm?

·       How you define the weight v of the different costs components?

Other points:

·       PG 1: IPPS define the acronym

·       PG 2: On -> on

·       English has to be revised.

I suggest a major revision of the paper before the publication.

Author Response

  1. In relation to the Literature described, it is necessary to underline the novelty of the paper; I suggest introducing a comparison table where, by different characteristics, the present work is compared with the Literature.

Respond: Thank you for your kind comment. We agree with you and have created a table to describe the related references in the revised manuscript and below:

References

Objective functions

Scheduling problems

Methodologies

Mohapatra et al [9]

Makespan,Idle time of machines,Machining cost

IPPS

NSGA

Zhang et al [10]

Makespan

IPPS

OCGA

Xia et al [11]

Makespan

IPPS

GAVNS

Zhao et al [14]

Makespan,Machining cost

IPPS

ACO

Mohamad et al [15]

Makespan,Machining cost

IPPS

NSGA(ELECTRE)

Chaudhry [16]

/

IPPS

GA

This work

Makespan,Machining time,Machining cost

IPPS

HBMO

  1. It is not clear how the aerospace market/components production changes the problem analyzed (for example in comparison with the automotive sector). I suggest explaining in the introduction.

Respond: Thank you for your kind comment. We agree with you and have explained how the aerospace market/components production changes the problem analyzed in the introduction. The detailed descriptions are available in the revised manuscript and below:

In this paper, we focus on an IPPS problem of complex aerospace components. Beacuse complex aerospace components are mixed with small batches, complex geometric features and changeable process routes, which lead to poor matching between processing features and manufacturing resources, the aerospace market can be used as a excellent IPPS case study.

  1. I suggest introducing in the appendix the sensitivity analysis for the different heuristic parameters.

Respond: Thank you for your good comment.We agree with you and have introduced a sensitivity analysis for the different heuristic parameters in the appendix. The detailed descriptions of the sensitivity analysis are given in the revised manuscript and below:

  1. I suggest analyzing the convergence capacity of the bee algorithm with and without the greedy rule (and the related importance of the greedy rule) in comparison with the NSGA-II algoritm.

Respond: Thank you for your comment.We agree with you and have analyzed the convergence capacity of the bee algorithm with and without the greedy rule (and the related importance of the greedy rule) in comparison with the NSGA-II algoritm below.

In this work, a multi-objective IPPS problem of aerospace complex components is addressed. Then, a HBMO is designed to solve the model. To best of our knowledge, greedy rule is the major component to guarantee the convergence capacity of the swarm and evolutionary algorithms. In this way, the algorithm with greedy rule is definitely superior to the algorithm without greedy rule. For the proposed improved HBMO and NSGA-II algorithm, the non-dominated sort strategy and external archive set are introduced to ensure the convergence capacity. The specific description of the combination of the non-dominated sort strategy and external archive set are summarized as follows:

( 1 ) Let i = 1 ;

( 2 ) For all j = 1, 2....n, and j ≠ i, according to the above definition, comparing the dominance and non-dominance between individuals and individuals ;

( 3 ) If no individual is superior, it is marked as non-dominated individual ;

( 4 ) Let i = i + 1, go to step ( 2 ) until all non-dominated individuals are found.

Moreover the non-dominated individual set obtained by the above steps is the first-level non-dominated layer of the population. Then, ignoring these marked non-dominated individuals ( that is, these individuals are no longer compared in the next round ), and then following steps ( 1 ) - ( 4 ), the second-level non-dominated layer is obtained. And so on until the whole population is stratified.

To obtain the crowding degree estimation of the solutions around the specific solution in the population, we calculate the average distance between two points on both sides of this point according to each objective function. This value is used as an estimate of the perimeter of the cuboid with the nearest neighbor as the vertex.The calculation of crowding degree ensures population diversity.

In addition, the external archive set is used to store the solutions obtained from each iteration of the proposed improved HBMO and NSGA-II algorithm. And the archive set is updated by replace the inferior solution with a better solution obtained from the searching process.

  1. If the problem size is small (only 6 parts), is it possible to generate the optimal solution with an exhaustive method in order to evaluate the convergence to the optimality of the proposed algorithm? 

Respond: Thank you for your comment. The traditional IPPS problems have been proven to be NP-hard, which can not be solved in the polynomial time. As an extend of the traditional IPPS problem, the presented multi-objective IPPS problem of aerospace complex components should also be NP-hard. In this way, the exhaustive methods will show low convergence and long optimization time. Thus, the proposed improved bee algorithm will definitely performs better performance than the exhaustive methods on convergence ability.

  1. The problem size considered is really very small (6 parts); it is possible to increase the number of parts in order to evaluate the performance of the algorithm?

Respond: Thank you for your good comment. We agree with you and have increased the number of parts in order to evaluate the performance of the algorithm in the revised manuscript and below.

Instance

HBMO(Improved)

NSGA-II

HBMO

Mean

Std

Mean

Std

Mean

Std

P8_F45_O2_W2_M3_T5

0.00723

0.00390

0.02563

0.02549

0.393622

0.12829

P10_F53_O2_W2_M4_T4

0.02669

0.01245

0.04486

0.02674

0.30907

0.16658

P15_F83_O2_W2_M4_T5

0.02695

0.01042

0.02827

0.01357

0.47791

0.24043

P25_F140_O2_W2_M3_T5

0.02197

0.01278

0.03438

0.01765

0.32007

0.12018

P40_F217_O2_W2_M4_T4

0.00705

0.00233

0.01800

0.01710

0.47586

0.21803

  1. How you define the weight v of the different costs components?

Thank you for your good comment. We have explained this reason of designing the weight of each target in the revised manuscript. The weight reflects the preference of the decision makers on time-saving, and cost-effective. In this work, we study an IPPS problem of aerospace complex components. Due to the complex production process of aerospace complex components, unreasonable production scheduling will lead to a lot of unnecessary production cost waste. Secondly, as a military product with strict planning nature, hybrid production of aerospace complex components needs to meet the requirements of on-time delivery. Thus, compared with cost optimization, we prefer to achieve operation time and cost optimization, followed by makespan optimization, then we set the weights of makespan, operation time and processing cost to be 0.5, 0.3, 0.2, respectively.

  1. Other points:
  • PG 1: IPPS define the acronym
  • PG 2: On -> on
  • English has to be revised.

Respond: Thank you for your good comment. We agree with you and have modified the issues such as below:

  • The definition of IPPS have been given.
  • “On”---->“on”
  • English writing has been revised carefully.

Reviewer 2 Report

In the paper, the authors propose an integrated optimization model for process planning and production scheduling of aerospace complex components with the optimization objectives of minimizing the maximum completion time, minimizing the total processing time of machine tools and minimizing the processing cost. The fact that deserves attention is that it the paper the authors consider multi-criteria scheduling issue.

The paper is properly organized and includes all important elements: introduction, related work discussion, detailed description of the proposed solution, results of the experiments, conclusions. The paper was very carefully prepared. As a consequence, the quality of the paper is high.

However, I have some minor remarks:

1. The term "Integrated process planning and scheduling" should be written as a "Integrated Process Planning and Scheduling".

2. In Figure 12 it is not clear which part is a), b), c)...

Other remarks have been marked into the attached pdf file.

Author Response

  1. The term "Integrated process planning and scheduling" should be written as a "Integrated Process Planning and Scheduling".

Respond: Thank you for your kind comment. We have made a correction here, and IPPS is defined in the appropriate position.

  1. In Figure 12 it is not clear which part is a), b), c)...

Respond: Thank you for your kind comment. We have re-modified the annotation of Figure 12.

Other remarks have been marked into the attached pdf file.

Respond: Thank you for your kind comment. We have made point-to-point modifications to the notes in the PDF file.In the first mark may be a journal typesetting problems lead to typesetting errors, in the original word we provide the place did not appear this problem.We have made changes about the figure number reference in the text.

Reviewer 3 Report

This paper introduced an integrated optimization method for complex component process planning and workshop scheduling. In the environment of aerospace manufacturing, there are some differences and challenges compared to general manufacturing, such as complex geometric features, small batches, changeable process routes, dynamic model requirements, etc. To handle these challenges, a multi-objective IPPS (integrated process planning and scheduling) model is established. The model aims to minimize the makespan, total processing time, and processing cost of the manufacturing process.  Further, to solve the model, they Proposed an improved bee mating algorithm based on four-layer encoding and five worker bee breeding strategies to solve the model, resulting in an efficient process planning and shop scheduling scheme that improves production efficiency while reducing production cost and machine load.

In my opinion, the topic of this study is very interesting, and of great importance in the real-world application. However, I doubt whether this paper is ready for publication, because it is poorly typesetted. Especially for the equations, the font size of some equations are large, and others are small. The figure and its caption are not in the same page. The English writting can be greatly improved, and there are many grammer mistakes and typos.

Author Response

In my opinion, the topic of this study is very interesting, and of great importance in the real-world application. However, I doubt whether this paper is ready for publication, because it is poorly typesetted. Especially for the equations, the font size of some equations are large, and others are small. The figure and its caption are not in the same page. The English writting can be greatly improved, and there are many grammer mistakes and typos.

Respond: Thank you for your kind comment. In view of the layout, language and other issues of the article, we have made corresponding processing. And please see the attachment.

Round 2

Reviewer 1 Report

thanks for your answers

no other comments

best regards

Author Response

Thank you very much for your opinion, thank you !